# Competencies of health personnel for the practice of health literacy in Brazil: A Delphi consensus survey

**Flaviane Cristina Rocha Cesar**[1,2☯]*, **Thassara Felipe de Sousa**[1‡], **Angela Gilda Alves**[2‡], **Katarinne Lima Moraes**[3‡], **Maria Alves Barbosa**[4‡], **Lizete Malagoni de Almeida Cavalcante Oliveira**[4☯]

**1** Faculdade de Medicina, Centro Universitário de Mineiros–UNIFIMES, Trindade, Goiás, Brazil, **2** Faculdade de Enfermagem, Centro Universitário UNIFASAM, Goiânia, Goiás, Brazil, **3** Faculdade de Ceilândia, Universidade de Brasília, Brasília, Distrito Federal, Brazil, **4** Faculdade de Enfermagem, Universidade Federal de Goiás, Goiânia, Goiás, Brazil

☯ These authors contributed equally to this work.
‡ TFS, AGA, KLM and MAB also contributed equally to this work.
\* flaviane_rocha01@hotmail.com

**Data Availability Statement:** All files are available from the figshare database (accession number(s) https://doi.org/10.6084/m9.figshare.19919528).

## Abstract

### Objective

The aim of this study was to identify a set of competencies of health personnel for the practice of health literacy in Brazil.

### Methods

Scoping review and online interviews with healthcare practitioners, followed by three rounds of the modified e-Delphi method with health literacy specialists from November/2020 to March/2021. During the rounds, the items were revised, new items added for review, and their importance was rated on a five-point Likert scale in an online form. Those items that achieved a mean Likert rating of 4+ (rated important to very important) and $\geq$ 90.0% agreement among the experts were maintained in each round.

### Results

The initial competencies list contained 30 items from the literature scoping review and online interview with 46 Brazilian healthcare practitioners. 25 experts (health personnel with publications on health literacy) were invited to participate in the e-Delphi rounds. Of the total of 56 items evaluated, 28 reached consensus among the experts. The Brazilian competencies list differed from other consensuses by the emphasis on professional commitment to the literacy in health, autonomy and social context of the patient.

### Conclusion

For the Brazilian context, 28 competencies are relevant to the practice of health literacy in health care. This study is an initial step to develop the HL competences of Brazilian health professionals and an update of the skills evidenced in previous international studies.

**Funding:** This study was partially supported by the Fundação De Apoio à Pesquisa Do Distrito Federal (FAPDF). No additional external funding was received for this study. The funder had no role in study design, data collection and analysis, decision to publish, or preparation of the manuscript.

**Competing interests:** The authors have declared that no competing interests exist.

## Introduction

Health literacy (HL) is defined as competences to access, understand, evaluate and use health information and services in order to make decisions for health promotion [1]. HL is more than a mental phenomenon or a set of skills, it must be understood as a set of social practices inserted in a given context [2].

Satisfying the needs of health service users is a challenge for healthcare personnel (HCP), considering the high prevalence of inadequate HL in the population of different countries. Studies suggest that 50.0 to 90.0% of the European, North American and Asian population have insufficient HL [3–5]. In Brazil, studies with specific populations, such as those with chronic diseases and the elderly, estimate that 45.4% to 66.0% of the population have inadequate HL [6–9].

Inadequate HL has important implications for well-being and has been associated with increased risk of death [10], non-adherence to medication [11], poor quality of life [12], less control of chronic diseases [13] and increased hospital readmission [14]. Therefore, the HCP need to assume as a universal principle the addition of health care models that incorporate the HL as a public health issue and quality of care.

Professional training for HL has been associated with the development of knowledge, skills and attitudes that enhance the effective response of professionals to the needs of patients with low HL [15]. Thus, professional competence is a starting point for literate health teams.

The first consensus on professional competences in HL was proposed by Coleman, Hudson and Maine [16]. The authors used a literature review and a panel of North American experts to establish a set of competencies in HL for HCP. Subsequent studies showed that most of these listed competencies could be applicable in European countries [17], Chinese [18] or in specific professions, such as nursing [19]. However, the removal and addition of items that occurred in these studies made clear the need to reapply and adapt the skills proposed by the original instrument to other places and cultures.

The establishment of consensus on HL competencies is supported by the new roles expected for HCP as health promoters in clinical settings, as professionals and researchers, according to the Shanghai Declaration on Health Promotion [20]. This declaration focuses on promoting HL, linking the capacity of individuals and communities, as well as the capacity of professionals and health systems to respond to this demand. In addition, the theme is aligned with the need for research in communication and health information provided for in the Agenda of Research Priorities in Brazil [21].

Currently, consensus on HCP competencies in HL is restricted to the European continent [17], North American [16] and Asia [18]. The lack of a model of competence in HL for Latin American countries like Brazil is a barrier to the construction of curricula for the formation of HCP and for the permanent education of professionals in HL. Therefore, this study aimed to identify a set of competencies of HCP for the practice of HL in Brazil.

## Methods

We use a modified e-Delphi study [22] composed of two stages: 1) development of a preliminary list of HL competencies through literature review [23] and online interviews with health professionals; 2) establishment of consensus through three evaluation rounds with HL experts. All competences developed in the study were written in Portuguese and only at the time of publication of the manuscript did we translate it into American English. We had co-authors with English (KLM) and Spanish (MAB) language experience during this process.

## A scoping review of domains of professional competencies in HL

The first stage of the study was a scoping review of the literature that was published in May 2022 [23]. We searched Medline (PubMed), CINAHL (EBSCO), PsycInfo, ERIC (ProQuest), Lilacs (BVS), and EMBASE (Elsevier) for original studies and documents in April 2020. We combined the descriptors "Health literacy" AND (Competence OR "Health Personnel") and similar ones in each database. Articles published in English, Spanish and Portuguese were included, regardless of the year of publication. The selected publications should address the topic of health literacy among health professionals in the care or academic environment, including: knowledge: studies that evaluated or described predominantly cognitive aspects of HCP on HL; Skills: studies that evaluated or described actions of HCP considering the patient's HL, in clinical practice or mental activities that stimulate critical thinking; Attitudes: studies that evaluated or described preferences, values and attitudes of HCP in relation to patients' HL.

## Interviews with healthcare professionals

The results of the scoping review were used to guide the content analysis of the interviews. We identified several competencies in health literacy through the scoping review. These were semantically approximated and repeated competencies were excluded. Thus, the items identified in the literature served to confirm competences with the specialists in national health literacy, allowing for additions.

We considered experts in health literacy to be professionals who had at least six months of experience in a care activity that involved carrying out educational activities with the patient. This time of experience in care can be considered minimal to allow the opportunity to experience HL practices, as shown by the result of a previous study [24].

The experts in health literacy were identified through e-mail accessed in health services, educational institutions and scientific publications on health literacy. Snowball sampling was used to identify other eligible participants by requesting the e-mail indication of reference professionals in patient health education in the survey form.

The process of recruiting participants and conducting the interviews continued until the inclusion of professionals from different categories and regions of the country was verified, and new categories did not enter the data set. The interviews were carried out from March to July 2020.

Individual online interviews lasting 15 to 40 minutes were conducted with 46 health professionals who worked in health services in Brazil. All interviews were conducted by the first author. All study authors have experience in qualitative research and health literacy and contributed to the interpretation of data.

## Operationalization of modified e-Delphi method rounds

The competencies identified in the literature review and interviews constituted the previous list that was submitted to the consensus of Brazilian specialists in HL.

Specialists in HL were identified considering the criterion of authorship in books or peer-review articles, this strategy has been used in studies with the Delphi method, indicating mastery of the topic [25].

The identification of participants for this committee of experts was carried out through a search with the term "Health Literacy" and a filter for the region of Brazil, in the last five years, in PubMed, Web of Science and LILACS. The search identified significant authors from different regions of the country, totaling 83 researchers from different professional categories in the health area publishing on HL in the Brazilian context.

All identified HL specialist researchers were invited to participate in the research (n = 83). Subsequently, the specialists who returned the invitation e-mail were submitted to the following inclusion criteria: having a degree in the health area; have experience in direct patient, family or community care, have at least one HL publication in the last five years. Exclusion criteria were being in function deviation; being away from work activities for more than five years; have exercised only administrative function in the health area. The characteristics of the specialists are described in Table 1.

The modified e-Delphi method consisted of three rounds of data collection *by e-mail*, each of which had varied duration, round 1 from 24.11.20 to 07.01.21; round 2 from 16.01.21 to 09.02.21 and round 3 from 15.02.21 to 10.03.21, making a period of 100 days between the months of November 2020 and March 2021. During Delphi, the number of participants varied in round 1 = 25, 100.0%; round 2 = 16, 64.0% and round 3 = 12, 48.0%.

In each round, the experts were invited to evaluate the importance of each question using a five-point Likert scale, ranging from 1 (not important) to 5 (very important), according to a previous study [18].

The experts received feedback and a summary of the results of each round, and their modified individual items were color-coded to avoid misunderstandings while reading in the second and third rounds.

**Table 1. Characterization of health literacy specialists (n = 25).** Brazil, 2021.

| Variable | | n | % |
|---|---|---|---|
| Sex | Female | 21 | 84.0 |
| | Male | 04 | 16.0 |
| Age (in years) | Less than or equal to 30 | 04 | 16.0 |
| | Between 31 and 40 | 11 | 44.0 |
| | Greater than 40 | 10 | 40.0 |
| Region of the country | North | 01 | 4.0 |
| | Northeast | 05 | 20.0 |
| | Midwest | 03 | 12.0 |
| | Southeast | 08 | 32.0 |
| | South | 08 | 32.0 |
| Academic level | Specialization | 01 | 4.0 |
| | Masters | 06 | 24.0 |
| | Doctorate | 18 | 72.0 |
| Type of service | Public | 18 | 72.0 |
| | Private | 04 | 16.0 |
| | Both | 03 | 12.0 |
| Professional category | Nurse | 11 | 44.0 |
| | Dentist | 05 | 20.0 |
| | Pharmacist | 03 | 12.0 |
| | Audiologist | 02 | 8.0 |
| | Physical educator | 01 | 4.0 |
| | Physical therapist | 01 | 4.0 |
| | Nutritionist | 01 | 4.0 |
| | Doctor | 01 | 4.0 |
| Time working on direct patient care (in years) | Less than or equal to 5 | 10 | 40.0 |
| | Between 6 to 10 | 03 | 12.0 |
| | Greater than 10 | 12 | 48.0 |

In statements for which consensus was not reached, feedback and comments from participants were used to adjust the statement for the next round. It has been suggested that experts fill out the form preferably within 15 days and reminders of the activity were sent every five days by email.

In round 3, the data were reorganized, giving rise to consensus on skills in HL for Brazilian health professionals.

### Data analysis

The transcripts of the interviews with HCP were submitted to thematic content analysis proposed by Bardin [26] with the help of Atlas Ti software. This framework allowed the identification of the corpus of analysis, through the recording units (excerpts from the participants' speeches). The registration units were approximated by similarity, resulting in initial categories in Atlas Ti. Subsequently, thematic grouping was performed, resulting in intermediate categories or subthemes. These sub-themes originated the themes, they were approximated and resulted in the final categories, presented here as competences.

Data from the e-Delphi rounds were analyzed using the statistical software Statistical Package for the Social Sciences (SPSS) version 20. Initially, a descriptive analysis of the sociodemographic variables and the list of competencies was performed using percentage [27].

Items suggested by participants in rounds 1 and 2 of the modified e-Delphi method were included in the Brazilian version if they reached $\geq$ 90% agreement among participants [7]. There is no consensus on the standard for the level of agreement between authors in the e-Delphi methodology, with a variation of 51.0% - 80.0% in the literature [6]. The cut-off point of 90.0% allowed: increasing the sensitivity of the participants' choice, reducing the risk of entering redundant items and increasing the possibility of including items that are really relevant and feasible to be applied to reality.

Items excluded for not reaching 90.0% were checked and approximated by their semantic similarity in order to confirm whether they would have their correspondents represented in the final list. Thus, it was possible to ensure that the 90.0% cut-off point did not exclude relevant items.

The Brazilian consensus was compared by the authors to the three previous consensuses, American, European and Chinese, through the evaluation of the INESC-ID@ASSIN system to identify semantic similarity on a Likert scale divided into: 1. Completely different, on different subjects; 2. Not related, but more or less on the same subject; 3. Something related. They may describe different facts, but share some details; 4. Strongly related, but some details are different; 5. Essentially the same thing [28].

### Ethics statement

The project was approved by the Research Ethics Committee of the Federal University of Goiás (CAEE registration number: 17226919.10000.5083). The right of refusal free of damages and the confidentiality of the identity of all participants were guaranteed.

## Results

Nurses (n = 16) and physicians (n = 13) constituted the majority of respondents, 58.0% (n = 17) of the participants had a time greater than or equal to 10 years of experience in patient care and health literacy had already been heard by 41.3% (n = 19) or studied during the master's or doctorate by 8.6% (n = 4) of the participants (Table 1).

Eighteen professional competences were identified for SL practice based on the synthesis of the content of 34 articles. The categorization of the content of the interviews showed 12 professional competencies in health literacy.

The initial competency list contained 30 items distributed in knowledge (5 items), skills (16 items) and attitudes (9 items), resulting from the literature review and interviews. In the first round, 22 items were added and in the second round, three items were added. However, of the total of 56 items evaluated, only 28 reached consensus among experts. Only items that achieved evaluation (3- important; 4- very important or 5- extremely important) in $\geq$ 90.0% of the evaluations in round 1 and/or 2 were evaluated in the third round (Table 2).

The experts attached great importance to the practical knowledge of the HL, encompassing its impact and low HL signals. The skills assessed by experts as important mainly included the aspect of adequacy of language and materials, learning verification and reinforcement strategies. In the context of attitudes, the experts chose items related to the expression of sense or conscience in HL, selecting items with terms of co-responsibility, sensitivity, empathy, respect and commitment. The assessment of the need for social support, the educative attitude and attitudes that make patients co-responsible in the educational process were differentiators of our study in relation to other lists of HCP competences in HL (Table 2).

## Discussion

Most of the participants in our study being nurses is consistent with surveys on the composition of health teams in the world, in which 59.0% of the workforce is composed of nurses [52]. In addition, studies suggest that nurses are leaders in building a health-literate society [53]. Thus, the expertise of this professional group brings to the analysis of the study an essential theoretical-practical component, without disregarding the multidisciplinary approach made possible by the inclusion of HL specialists from other professional categories.

The higher frequency of participants with a doctorate observed in our study may be associated with the fact that the HL is still timidly part of the national curricula [54]. It is likely that HCP have a greater chance of coming into contact with the concept of HL in the context of postgraduate studies. This hypothesis can be supported by a review that evidenced the expressive Brazilian scientific production in the master's and doctorate [55].

The assessment of the need for social support, the educative attitude and attitudes that make patients co-responsible in the educational process were differentials of our study in relation to other lists of HCP competences in HL. Previous studies have suggested that HL theoretical knowledge is an important competency, including concepts, definitions and guidelines [16, 17, 24]. Em contraponto, o conhecimento teórico do HL não foi priorizado pelos nossos especialistas refletindo uma proposta prática para os itens de competência.

The skills related to adaptations in HL practices present in the current study reflect phenomena such as globalization, migratory movements and social disparities. These result in idiomatic, cultural and social determinants challenges in the practice of HL. Recent studies have highlighted the relationship of HL in predicting health disparities among immigrants at the national level [56] and the mediating role of HL in the established social inequalities [57].

The ability to assess HL through instruments was excluded, probably because it is not very applicable to reality, despite being recommended in the literature [18], it may require a time of attention not available by professionals [33]. In addition, current instruments may have complex application or may not address cultural and clinical issues relevant to different care contexts. For example, most instruments are not available in Brazilian Portuguese, they make an indirect assessment through health behaviors and are limited to functional aspects of the HL [58].

**Table 2. Consensus of competencies of health personnel for the practice of health literacy in Brazil (n = 12).** Brazil, 2021.

| Competencies in health literacy (HL) practices | Source | Round accepted | Round final | Level of semantic similarity of the approved item with lists from other studies* | | |
|---|---|---|---|---|---|---|
| | | | Percentage of $\geq 4$** | USA[a] | Europe[b] | China[c] |
| **Knowledge** | | | | | | |
| Recognize at least one definition of HL | [16, 18, 29–33] | Excluded | 50.0 | - | - | - |
| Recognize the impact of HL on patient care | [16, 18, 29–33] | 2 | 100.0 | 4 | 4 | 4 |
| Identify low HL signs | [16, 18, 29, 34] | 1 | 91.7 | 5 | 5 | 5 |
| Know guidelines for assertive communication in health | [35–42] | Excluded | 83.3 | - | - | - |
| Know strategies to evaluate interventions in HL | [16, 18] | Excluded | 83.4 | - | - | - |
| Have knowledge about the subject to be taught | Interview | 3 | 91.6 | 1 | 1 | 1 |
| Know the patient's HL assessment tools | Round 2 specialists | Excluded | 83.3 | - | - | - |
| Know the difference between functional health literacy and literacy | Round 1 specialists | Excluded | - | - | - | - |
| Recognize people-centered care, health care models, health self-management, self-efficacy | Round 1 specialists | 2 | 91.7 | 1 | 1 | 1 |
| Recognize aspects of the HL of the patient's community/context | Round 1 specialists | Excluded | - | - | - | - |
| Recognize ease and difficulty of access and whether the material allows or favors understanding of the topic addressed | Round 1 specialists | 2 | 100.0 | 4 | 4 | 3 |
| Consider human freedom of choice | Round 1 specialists | Excluded | - | - | - | - |
| Recognize whether the patient or person being health literate can make healthy or unhealthy choices. | Round 1 specialists | Excluded | 83.3 | - | - | - |
| Recognize whether the patient or person needs governmental social support to put what they have learned into practice | Round 1 specialists | 2 | 91.6 | 3 | 3 | 1 |
| Knowing learning styles for interventions in professionals with limited HL | Round 1 specialists | Excluded | - | - | - | - |
| **Skills** | | | | | | |
| Use simple language in the transmission of health information | Round 1 specialists | 3 | 100.0 | 5 | 5 | 5 |
| Identify the need to adapt the conduct and learning materials to the patient's health literacy level | [16, 18] | 1 | 100.0 | 4 | 4 | 4 |
| Develop and adapt educational materials to each target audience according to HL fundamentals | Round 1 specialists | 2 | 100.0 | 4 | 4 | 5 |
| Assess the level of health literacy of patients | [16, 36, 38, 43–45] | Excluded | 75.0 | - | - | - |
| Use strategies to reinforce patient learning in health | [16, 18, 36, 40, 43, 45–49] | 2 | 100.0 | 4 | 4 | 4 |
| Check patient learning in health | [16, 18, 36, 40, 43, 45–49] | 2 | 100.0 | 5 | 5 | 5 |
| Use the health literacy assessment to plan strategies appropriate to each individual's level of sufficiency | Round 2 specialists | Excluded | 83.3 | - | - | - |
| Apply health patient education strategies designed with the assumptions of health literacy | [16, 18] | 3 | 91.6 | 4 | 4 | 4 |
| Develop communication focused on health literacy through dialogue, simple language, cultural/regional terms, imagery language, eye contact and teaching materials | Interview | 1 | 100.0 | 4 | 4 | 4 |
| Guide and enable clients for health self-management | [39, 50, 51] | 1 | 91.6 | 1 | 1 | 5 |
| Involve patients in the consolidation of their health rights and care plan | Interview | 1 | 100.0 | 1 | 1 | 1 |
| Build shared decision-making and relationship with the patient/family/caregiver | [16, 18, 34, 38, 46, 50, 51] | 2 | 100.0 | 3 | 3 | 1 |

*(Continued)*

**Table 2.** (Continued)

| Competencies in health literacy (HL) practices | Source | Round accepted | Round final | Level of semantic similarity of the approved item with lists from other studies* | | |
|---|---|---|---|---|---|---|
| | | | Percentage of $\geq 4$** | USA[a] | Europe[b] | China[c] |
| Strengthen the individual's autonomy | Round 1 specialists | 2 | 100.0 | 1 | 1 | 1 |
| Build interprofessional collaboration through sharing and discussing cases with peers | [18] | Excluded | 75.0 | - | - | - |
| Evaluate the patient's biopsychosocial, emotional, educational, cultural and linguistic characteristics that may interfere with their teaching-learning process | Interview | Excluded | 83.3 | - | - | - |
| Associate educational content with the patient's reality | Interview | 1 | 100.0 | 5 | 5 | 5 |
| Identify social context, general and health literacy, usual patient knowledge sources and health determinants | Interview | 1 | 91.6 | 1 | 1 | 1 |
| Adapt the care plan and learning materials to the patient's biopsychosocial, emotional, educational, age group, cultural, linguistic and health literacy level | Interview | 1 | 100.0 | 4 | 4 | 4 |
| Use information and communication technology whenever possible to assist in the process of teaching patients | Interview | 1 | 91.6 | 4 | 4 | 4 |
| Teach the patient to access reliable health information | Round 1 specialists | 3 | 91.6 | 1 | 1 | 1 |
| Implement strategies to promote greater health literacy and learning among patients | Round 1 specialists | Excluded | 83.3 | - | - | - |
| Promote patient empowerment | Round 1 specialists | Excluded | 83.3 | - | - | - |
| Carry out a diagnosis of the territory of insertion of the actions | Round 1 specialists | Excluded | 75.0 | - | - | - |
| Perform diagnosis of the support network | Round 1 specialists | Excluded | 75.0 | - | - | - |
| Work with pounds if necessary | Round 1 specialists | Excluded | 66.6 | - | - | - |
| Involve social and family support network in the therapeutic plan | Round 1 specialists | Excluded | 75.0 | - | - | - |
| Identify the patient's feelings and emotions that can interfere with the teaching process | Round 1 specialists | Excluded | 83.3 | - | - | - |
| Use strategies that suit both insufficient and sufficient health literacy | Round 1 specialists | Excluded | - | - | - | - |
| **Atitudes** | | | | | | |
| Demonstrate intent and confidence in using health literacy skills | [47, 48, 51] | Excluded | 66.6 | - | - | - |
| Demonstrate having become an agent of change in health literacy | [35] | Excluded | 58.3 | - | - | - |
| Demonstrate changing perspectives, assumptions and expectations as a result of health literacy actions | [35] | Excluded | - | - | - | - |
| Feeling co-responsible for the patient's health literacy | [42] | 2 | 91.6 | 4 | 4 | 1 |
| Be sensitive and empathetic to patients' unsuccessful experiences in the healthcare system | Round 2 specialists | 3 | 91.6 | 3 | 3 | 1 |
| Feeling responsible for taking care of patients' communication needs | [16] | 1 | 91.6 | 5 | 5 | 1 |
| Demonstrate a respectful and non-critical attitude towards individuals with limited health literacy skills | [16] | 1 | 100.0 | 5 | 5 | 4 |
| Demonstrate continuous learning intent in health literacy practices or willingness to learn | Interview | Excluded | 50.0 | - | - | - |
| Demonstrate an educational attitude, expressing liking what you do, being committed, dynamism, proactivity, patience and the desire to help the patient to develop their health literacy | Interview | 1 | 100.0 | 1 | 1 | 1 |
| Manage your emotions during the educational process, such as anguish, ego, frustrations | Interview | Excluded | 83.3 | - | - | - |

(Continued)

**Table 2.** (Continued)

| Competencies in health literacy (HL) practices | Source | Round accepted | Round final | Level of semantic similarity of the approved item with lists from other studies[*] | | |
|---|---|---|---|---|---|---|
| | | | Percentage of $\geq 4$[**] | USA[a] | Europe[b] | China[c] |
| Seeking the patient's commitment to health care, without blaming him, but trying to make him co-responsible. | Round 1 specialists | 2 | 100.0 | 1 | 1 | 1 |
| Demonstrate self-efficacy, compassion, empathy, motivation and control | Round 1 specialists | Excluded | - | - | - | - |
| Ability to recognize the patient | Round 1 specialists | Excluded | - | - | - | - |

Note

[*]Sistema INESC-ID@ASSIN [28]—- 1. Completely different, on different subjects; 2. Not related, but more or less on the same subject; 3. Something related. They may describe different facts, but share some details; 4. Strongly related, but some details are different; 5. Essentially the same thing.

[**]percentage of experts who assigned a rating greater than or equal to four for the item's level of importance.

[abc]References A- Coleman, Hudson (16); B- Karuranga, Sørensen (17); C- Chang, Chen (18).

In this sense, universal precaution presupposes that health information is offered in a simple way and that health services are organized and accessible to their users, regardless of the HL level of individuals [32, 59]. Our experts, in line with this perspective, have understood that assessment skills are less important than the application or use of HL-based strategies.

In the context of attitudes, items related to empathy, responsibility, respect and commitment reached consensus. Understanding these items as important is associated with the emotional and relational characteristic of the attitude domain [60]. According to Perrenoud [60], this domain of competences is characterized by values and principles, following a subjective perspective and intrinsically linked with knowledge and action.

Our study differed from the other consensuses by developing a list that provides a transversal approach to professional competence in HL, with a view to proposing minimum components for the formation of HCP from graduation to professional training. In addition, the social context, favoring patient autonomy and professional commitment were points that our study identified and that had not been evidenced in other studies on HL competencies [16–18].

Although the Delphi method is recognized for allowing consensus, some limitations of this study need to be considered in our results. The limitations of the study included the intentional sampling and the greater number of nurses, doctors and civil servants that may restrict the application of competences in scenarios with different professional profiles. Although the scoping literature review and heterogeneous sampling may have allowed for a large number of items.

## Conclusions

The Brazilian consensus resulted in 28 items distributed in knowledge, skills and attitudes for the practice of HL by HCP. Although the consensus has been established to the practice of HL in Brazil, the items included in the Brazilian version may reveal important aspects for HCP in other countries, such as the assessment of the patient's social context and share responsibility for the educational process with patients (Table 2).

This study is an initial step to develop the HL competences of Brazilian health professionals and an update of the skills evidenced in previous international studies [16, 18].

## Author Contributions

**Conceptualization:** Flaviane Cristina Rocha Cesar, Thassara Felipe de Sousa, Angela Gilda Alves, Lizete Malagoni de Almeida Cavalcante Oliveira.

**Data curation:** Flaviane Cristina Rocha Cesar, Lizete Malagoni de Almeida Cavalcante Oliveira.

**Formal analysis:** Flaviane Cristina Rocha Cesar, Thassara Felipe de Sousa, Lizete Malagoni de Almeida Cavalcante Oliveira.

**Investigation:** Lizete Malagoni de Almeida Cavalcante Oliveira.

**Methodology:** Flaviane Cristina Rocha Cesar, Lizete Malagoni de Almeida Cavalcante Oliveira.

**Supervision:** Lizete Malagoni de Almeida Cavalcante Oliveira.

**Validation:** Angela Gilda Alves, Katarinne Lima Moraes.

**Visualization:** Thassara Felipe de Sousa, Angela Gilda Alves, Katarinne Lima Moraes, Maria Alves Barbosa, Lizete Malagoni de Almeida Cavalcante Oliveira.

**Writing – original draft:** Flaviane Cristina Rocha Cesar, Thassara Felipe de Sousa, Angela Gilda Alves, Katarinne Lima Moraes, Maria Alves Barbosa, Lizete Malagoni de Almeida Cavalcante Oliveira.

**Writing – review & editing:** Flaviane Cristina Rocha Cesar, Thassara Felipe de Sousa, Angela Gilda Alves, Katarinne Lima Moraes, Maria Alves Barbosa, Lizete Malagoni de Almeida Cavalcante Oliveira.

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
