## [Decision Letter · Decision Letter 0]

18 Mar 2022

PONE-D-22-06015COMPETENCIES OF HEALTH PERSONNEL FOR THE PRACTICE OF HEALTH LITERACY IN BRAZIL: A DELPHI CONSENSUS SURVEYPLOS ONE

Dear Dr. Cesar,

Thank you for submitting your manuscript to PLOS ONE. After careful consideration, we feel that it has merit but does not fully meet PLOS ONE’s publication criteria as it currently stands. Therefore, we invite you to submit a revised version of the manuscript that addresses the points raised during the review process.

We look forward to receiving your revised manuscript.

Kind regards,

Amene Abebe Kerbo, Ph.D.

Academic Editor

PLOS ONE

Journal Requirements:

3. Please include a copy of the PRISMA-Scr checklist and flow chart in support of the scoping review aspect of the study.

Additional Editor Comments:

Dear authors Thank you for your attempt to address a very nice area of research. However, you need to work harder before you resubmit the corrected version. Almost all of the reviewers comments are expected to addressed.

Reviewers' comments:

Reviewer's Responses to Questions

**Comments to the Author**

1. Is the manuscript technically sound, and do the data support the conclusions?

Reviewer #1: Partly

Reviewer #2: Yes

2. Has the statistical analysis been performed appropriately and rigorously? 

Reviewer #1: Yes

Reviewer #2: Yes

3. Have the authors made all data underlying the findings in their manuscript fully available?

Reviewer #1: No

Reviewer #2: Yes

4. Is the manuscript presented in an intelligible fashion and written in standard English?

Reviewer #1: No

Reviewer #2: Yes

5. Review Comments to the Author

Reviewer #1: Comments to the Authors

Thank you very much for giving me the chance to review your manuscript. I have included the following general comments:

The topic is very interesting and current issue on the health literacy of health professionals since the quality of the health system a little beat lowered. But this study is best used as an initial step to develop or update the healthcare competencies before being incorporated into the curricula. Because developing a competency may necessitate addressing a variety of factors such as healthcare provider and patient interaction, administration, personal factors, psychosocial factors, behavioral and cultural factors or indicators, and so on in order to be included in curricula and training tools, So the authors would do well to modify the conclusion part and also address the limitations of this study. The authors also need to correct the grammatical errors.

Introduction part

In lines 42 and 43, what is the abbreviation LS and SL stands for, respectively? Please try to write in detail if these were used for the first time in your manuscript.

Methods part

Articles published in English, Spanish, and Portuguese were included. How and in which language will the competencies be developed if this becomes real? How did you interview the expert? If there is language variation, how will it affect the standard? How did you control any bias that might exist because of such a difference?

The author used mean for judgment, but which measure of central tendency would be more appropriate: is that median or mean? This is because the experts’ consensus is 50%/50%.

The experts (health personnel with publications on health literacy) were selected based on their previous publications. Can you conclude that all of these are really experts? The agreement or consensus is also 50%. Again, they are selected online based on their publications. Do you think these publications are free from publication bias? Do you think that people who have many publications have expertise? Do you think that these health professionals represent the experts in Brazil?

The data was collected using an online interview. Was the data collector the author/s by himself/themselves? Is the problem due to competency or due to other factors?

How did the authors check the validity and reliability of the questionnaire to reach such a conclusion?

Table 1: Is the age category mutually exclusive? Again, the variable time working category too. Please make it clear. Percentage should be indicated with period “.” rather comma“,”

Conclusion part

The authors concluded that competencies should be embedded in health team training and academic curricula. Is it possible to incorporate and conclude in this manner based on a single finding because the issue is medial or a life issue, and embedding a program in curricula and providing training based on these competencies may necessitate additional steps such as including other stakeholders, findings, guidelines, principles, and discussions and others.

The authors considered experts in health literacy to be professionals who had at least six months of experience in a care activity that involved carrying out educational activities with the patient (line 82). Do you think these HPs are experts? Do you have a good understanding of healthcare literacy? Do you understand the healthcare system well?

Reviewer #2: Thank you very much for coming up with such an interesting topic. Your study might advance the health care delivery system by pointing problems related to health literacy. Saying this I have the following questions and comments…..

Introduction section line 40....the abbreviation LS should be defined in its first use...

Introduction section line 43....Define the abbreviation SL in its first use

Introduction section line 43....Inadequate SL has important implications for well-being and has been associated with increased risk of death hospital readmission....I am not clear with sentences...could you revise this sentences...I haven't understand it or it my be due to unknowing of the abbreviation SL....

Generally it’s better if you add the experiences of other countries competence in HL. Also, its better if you describe common competence's applied in various countries to convince reader about your topic of interest

Method section Line 70...you have included researches done in three languages (English, Spanish and Portuguese). Why only this three languages....?

Since you intention is to develop competencies in HL could you think that considering six month of professional experience is enough to select participants... Can we say that health professional with six months of experience as an expert. Can you define operationally what we mean an expert. Since the competence's you developed are going to be the part of curricula it needs high experts…

Its better if you move line 101-105 of method section to result part.

Method section line 166. Do you have any references to include >= 90% as a cut point.

In discussion section you used other language than english.....Em contraponto, o conhecimento teórico do HL não foi priorizado pelos nossos especialistas refletindo uma proposta prática para os itens decompetência. Please could you write it in English,,,as long as you used English as a primary language in this study.

6. PLOS authors have the option to publish the peer review history of their article (what does this mean?). If published, this will include your full peer review and any attached files.

Reviewer #1: **Yes: **Abiyot Wolie Asres

Reviewer #2: No

---

## [Author Response · Author response to Decision Letter 0]

6 Jun 2022

Dear Academic Editor and Reviewers,

We appreciate your feedback on our manuscript. We describe below the changes and explanations about our study in two parts, the first refers to the Academic Editor's comments and the second part contemplates the responses to the reviewers.

Part 1

We would like to thank you for your evaluation of our manuscript, we are delighted with your valuable suggestions. We inform you that to meet the requirements of the journal, we carry out a review of the manuscript, as described below.

Regarding the item 1: 

“1. Please ensure that your manuscript meets PLOS ONE's style requirements, including those for file naming. The PLOS ONE style templates can be found at 

https://journals.plos.org/plosone/s/file?id=ba62/PLOSOne_formatting_sample_title_authors_affiliations.pdf”

Authors' response: We reviewed the formatting of the entire manuscript according to the templates provided in the links above.

Regarding the item 2: 

Authors' response: We carry out the inclusion of the title page at the beginning of our manuscript.

Regarding the item 3: 

3. Please include a copy of the PRISMA-Scr checklist and flow chart in support of the scoping review aspect of the study.

Authors' response: 

Our scoping review was published in the journal Health literacy research and practice [1]. In this publication we have included Flow diagram for the scoping review process adapted from the PRISMA Extension for Scoping Reviews. We have included a copy of the PRISMA-Scr checklist to support our published scoping review [1]. We cite the reference in the method and explain how we use the article in the research. 

The first stage of the study was a scoping review of the literature that was published in May 2022 [1]. The detailed combination of search terms is described in the study protocol available at https://figshare.com/s/161960aa6503ee329208

Regarding the item 4: 

Authors' response: 

We would like to inform you that we do not have a repository of study data, so we have updated our Data Availability statement. Our data is saved in files in docx and xlsx format, in Portuguese. We can make them available if you request.

Part 2

Dear reviewers,

We would like to thank you for your important and careful analysis carried out on our manuscript. We inform you that we accept your suggestions for changes for our study. We review the manuscript, as described below:

Reviewer #1: Comments to the Authors

Thank you very much for giving me the chance to review your manuscript. I have included the following general comments:

The topic is very interesting and current issue on the health literacy of health professionals since the quality of the health system a little beat lowered. But this study is best used as an initial step to develop or update the healthcare competencies before being incorporated into the curricula. Because developing a competency may necessitate addressing a variety of factors such as healthcare provider and patient interaction, administration, personal factors, psychosocial factors, behavioral and cultural factors or indicators, and so on in order to be included in curricula and training tools, So the authors would do well to modify the conclusion part and also address the limitations of this study. The authors also need to correct the grammatical errors.

Authors' response: 

We followed reviewer #1's suggestion by changing the study's conclusion to: 

“The Brazilian consensus resulted in 28 items distributed in knowledge, skills and attitudes for the practice of HL by HCP. Although the consensus has been established to the practice of HL in Brazil, the items included in the Brazilian version may reveal important aspects for HCP in other countries, such as the assessment of the patient's social context and share responsibility for the educational process with patients (S2 Table).

This study is an initial step to develop the HL competences of Brazilian health professionals and an update of the skills evidenced in previous international studies [2, 3]”.

Reviewer #1: Comments to the Authors

Introduction part

In lines 42 and 43, what is the abbreviation LS and SL stands for, respectively? Please try to write in detail if these were used for the first time in your manuscript.

Authors' response: 

We fixed the typo. The two abbreviations refer to Health Literacy (HL).

Reviewer #1: Comments to the Authors

Methods part

Articles published in English, Spanish, and Portuguese were included. How and in which language will the competencies be developed if this becomes real? How did you interview the expert? If there is language variation, how will it affect the standard? How did you control any bias that might exist because of such a difference?

Authors' response: 

The items generated from the publications in English, Spanish and Portuguese resulted in a list of competencies in the Portuguese language. This list was presented to practical experts for correction, inclusion and adaptation to the Brazilian context.

 We have two types of specialists, practical specialists (health professionals with experience in patient education) and health literacy specialists (health professionals with experience in patient education who also have experience in studying and application of the term health literacy in their work). All interviews with practical experts were by video call by the lead author (FCRC) through the Google Meet app. The audio and image of all interviews were recorded. Health literacy experts evaluated the list resulting from the literature review and interview with health professionals in Brazil through an online form.

The interviews were conducted in Brazil in Portuguese. Data analysis also took place with reference to the Portuguese language. Only the manuscript was translated by the authors into the English language. The control of possible translation biases was controlled through the participation of co-authors in the construction of the competency list. We have co-authors with English (KLM) and Spanish (MAB) language experience during this process. In the manuscript, the changes suggested by the translation specialists were made. Co-author KLM has experience and publications on translation and cross-cultural adaptation of a health literacy instrument from English to Portuguese. Some KLM publications can be found at

https://www.scielo.br/j/ape/a/XC8t5yGWj7f78vLjt3QWRyL/abstract/?lang=en

https://dx.doi.org/10.1590%2F1518-8345.4362.3436

All competences developed in the study were written in Portuguese and only at the time of publication of the manuscript did we translate it into American English. 

Reviewer #1: Comments to the Authors

The author used mean for judgment, but which measure of central tendency would be more appropriate: is that median or mean? This is because the experts’ consensus is 50%/50%.

Authors' response: 

The experts' judgment was performed considering the analysis steps of the Delphi method and theoretical references related to the expert consensus [4]. Consensus on the e-Delphi method is commonly established by averaging participants' responses. The analysis of agreement using the percentage is in accordance with classical references of expert consensus [4]

Reviewer #1: Comments to the Authors

The experts (health personnel with publications on health literacy) were selected based on their previous publications. Can you conclude that all of these are really experts? The agreement or consensus is also 50%. Again, they are selected online based on their publications. Do you think these publications are free from publication bias? Do you think that people who have many publications have expertise? Do you think that these health professionals represent the experts in Brazil?

Authors' response:

As described in our manuscript in lines 131-139, the initial identification of health literacy specialists was performed using the criteria of scientific publications on health literacy in Brazil. We used the e-mail address of the corresponding author as a reference, who generally assumes greater responsibility for the published study. Furthermore, the authorship criterion is commonly used in Delphi studies. [4] and replicated in a study similar to ours [5].

In addition, as described in lines 140-147, participants who returned to the invitation email filled out an online form that allowed them to apply the inclusion criteria: having a degree in the health area; have experience in direct patient, family or community care, have at least one HL publication in the last five years. Exclusion criteria were being in function deviation; being away from work activities for more than five years; have exercised only administrative function in the health area.

Thus, in addition to the scientific publication criteria, we are concerned with identifying the experience of our specialists in health services. We have professionals from all regions of the country and 60.0% of the participants had more than six years of experience in care. A previous study showed that the time of six months of experience was enough for the specialists to be able to contribute to the list of competence in health literacy in the Chinese context [2].

Finally, the selection of our experts followed the principle of the power of information [6]. In accordance with this 'principle, our health literacy experts met the objectives of the study, contained specific knowledge about HL, the selection criteria were based on an established theory [4] and our analysis strategy considered the reflection of the group for the consideration of the items, reducing potential individual biases [7]. All participants marked the online form that they considered themselves to be experts in health literacy.

Reviewer #1: Comments to the Authors

The data was collected using an online interview. Was the data collector the author/s by himself/themselves? Is the problem due to competency or due to other factors?

How did the authors check the validity and reliability of the questionnaire to reach such a conclusion?

Authors' response:

The first author (Cesar, F.C.R) conducted all interviews alone. All interviews were audio and video recorded. The analysis of the interviews was carried out by the group of researchers, all of whom have experience with qualitative research.

Data quality was guaranteed considering the criteria proposed by Lincoln and Guba [8]: confirmability, credibility, dependability and transferability.

Reliability comprises the extent to which the study results express the participants' ideas and experiences [8]. For this, an analysis of the collection report in pairs was performed, where the authors examined the presence of preconceptions of the interviewer researcher, building an individual analysis with a level of understanding and agreement of the research team.

Credibility means how coherent the research results are and make sense to the people studied and the readers. It was carried out considering that the authors have different experiences and interests. Reading and discussing the interpretation among peers helps to identify preconceptions in the data and to identify whether there was any influence from the interviewer on the research.

Dependability means how consistent and stable research development is over its duration [8]. To meet this criterion, all materials related to the data and analysis of the study were collected in a comprehensive and chronological way, forming an operational trail that allows it to be audited.

Finally, transferability is the secure realization that the results of a study are applicable to other contexts [8]. In this sense, contextual information about the research results was described in as much detail as possible, so that readers could assess whether the results could be transferable or not. For this, we describe professional characteristics of the specialists interviewed, such as professional category, region of activity in the country and level of assistance in which they were inserted.

The perspective of validity and reliability of the questionnaire does not apply in the present study. We are building consensus among experts on a certain topic. For this, we followed the use of the Likert-type scale to classify the importance of the items. In each round, experts were invited to evaluate the importance of each question using a five-point Likert scale, ranging from 1 (not important) to 5 (very important), according to a previous study [2]. 

Reviewer #1: Comments to the Authors

Table 1: Is the age category mutually exclusive? Again, the variable time working category too. Please make it clear. Percentage should be indicated with period “.” rather comma“,”

Authors' response:

Yes, each category is mutually exclusive.

The suggested change to the percentage was made in the manuscript.

Reviewer #1: Comments to the Authors

Conclusion part

The authors concluded that competencies should be embedded in health team training and academic curricula. Is it possible to incorporate and conclude in this manner based on a single finding because the issue is medial or a life issue, and embedding a program in curricula and providing training based on these competencies may necessitate additional steps such as including other stakeholders, findings, guidelines, principles, and discussions and others.

Authors' response:

We have updated our conclusion to consider our result as an initial step towards the development of health literacy competencies in Brazil and as an update of international competencies.

Reviewer #1: Comments to the Authors

The authors considered experts in health literacy to be professionals who had at least six months of experience in a care activity that involved carrying out 

educational activities with the patient (line 82). Do you think these HPs are experts? Do you have a good understanding of healthcare literacy? Do you understand the healthcare system well?

Authors' response:

In addition to the scientific publication criteria, we are concerned with identifying the experience of our specialists in health services. We have professionals from all regions of the country and 60.0% of the participants had more than six years of experience in care. A previous study showed that the time of six months of experience was enough for the specialists to be able to contribute to the list of competence in health literacy in the Chinese context [2].

Finally, the selection of our experts followed the principle of the power of information [6]. In accordance with this principle, our health literacy experts met the objectives of the study, contained specific knowledge about SL, the selection criteria were based on an established theory [4] and our analysis strategy considered the reflection of the group for the consideration of the items, reducing potential individual biases [7]. All participants marked the online form that they considered themselves to be experts in health literacy.

To clarify our understanding of health literacy, we highlight that:

The lead author (F.C.R.C.) is part of the communication department of the Brazilian Health Literacy Network (REBRALS). Pioneer organization in Brazil to bring together researchers from different regions of the country who study health literacy. The researcher is a professor in the medical course at the University Centro Universitário de Mineiros (UNIFIMES) in the discipline of public health. The author developed her thesis on the topic of professional competences in health literacy to obtain her doctorate degree.

The third author (A.G.A.) is a Doctor in Nursing and is part of the Center for Studies in Assistance Paradigms and Quality of Life – NEPAQ at the Federal University of Goiás. In the city of Goiânia, Goiás, she worked as a supervising nurse at the Cidade Jardim Hospital and Maternity Hospital (1990-1993), director of the São Vicente de Paulo Nursing Technician and Assistant School (1992-1998), Nursing Manager (1994-2005) and general manager in the area of Psychiatry and Mental Health (2011-2016) at Clínica Isabela, and supervisory nurse in Psychiatry at Universidade Salgado de Oliveira (2007-2009). She served as special administrative supervisor (2014-2015) of the Municipal Health Department of Senador Canedo-GO. She worked as a teacher at the Pontifical Catholic University of Goiás (1999, 2002-2008), and at the Salgado de Oliveira University (2007-2009). 

The fourth author (K.L.M.) has a Doctorate in Nursing, experience in the area of Nursing, with an emphasis on Fundamental Nursing and Public Health. She works as a researcher mainly on topics related to health literacy, cross-cultural adaptation and validation of health measurement instruments; health education, non-communicable chronic diseases and quality of life. Member of the Center for Studies in Assistance Paradigms and Quality of Life (NEPAQ) and the Research Group on Health Promotion and Comprehensive Care (GIPIC) at the Pontifical Catholic University of Goiás. Current vice-coordinator of the Brazilian Health Literacy Network (REBRALS) and member of the International Health Literacy Association (IHLA).

Maria Alves Barbosa is a senior professor at the Faculty of Nursing at the Federal University of Goiás. She has a PhD in Nursing from the University of São Paulo (1994). She is currently a volunteer professor at the Postgraduate Program in Health Sciences at the Faculty of Medicine and at the Postgraduate Program in Nursing, both at the Federal University of Goiás, working on the following topics: Quality of Life, Research Ethics, Health of Worker, Complementary Therapies, Administration of Services and Nursing Assistance, Assistance Paradigms. She coordinates the Center for Studies in Assistance Paradigms and Quality of Life-NEPAQ.

Lizete Malagoni de Almeida Cavalcante Oliveira is a PhD in Health Sciences. Professor at UFG, assigned to the Faculty of Nursing (FEN) in 1983, being a Full Professor from 2014 to 2021. She was Coordinator of the Graduate Program in Nursing (PPGENF) at FEN/UFG from 2016 to 2020. Retired in 02/2021, she is a volunteer Professor at UFG, performing activities as Permanent Professor and Vice-Coordinator of PPGENF-FEN/UFG. Master's and Doctoral advisor at PPGENF-FEN/UFG, in the research lines "Theoretical, methodological and technological foundations for health and nursing care" and "Health and nursing management". Member of the Qualitative Health and Nursing Study Group (NEQUASE), of the International Health Literacy Association and of the Brazilian Health Literacy Network (REBRALS). She works in the areas of quality of life, health literacy, and emergency and critical care nursing care.

 Reviewer #2: Comments to the Authors

Introduction section line 40....the abbreviation LS should be defined in its first use...

Introduction section line 43....Define the abbreviation SL in its first use

Authors' response:

The correction was made in the manuscript.

SL and LS stand for health literacy. The correct abbreviation is HL and has been updated in the manuscript.

Reviewer #2: Comments to the Authors

Introduction section line 43....Inadequate SL has important implications for well-being and has been associated with increased risk of death hospital readmission....I am not clear with sentences...could you revise this sentences...I haven't understand it or it my be due to unknowing of the abbreviation SL....

Authors' response:

We rewrite the sentence to: Inadequate HL has been associated with increased risk of death [9], non-adherence to medication [10], poor quality of life [11], less control of chronic diseases [12] and increased hospital readmission [13]. Therefore, the HCP need to assume as a universal principle the addition of health care models that incorporate the HL as a public health issue and quality of care.

Reviewer #2: Comments to the Authors

Generally it’s better if you add the experiences of other countries competence in HL. Also, its better if you describe common competence's applied in various countries to convince reader about your topic of interest

Authors' response:

To address the suggestions, we have added the paragraphs below in the introduction:

The first consensus on professional competences in HL was proposed by Coleman, Hudson and Maine [3]. The authors used a literature review and a panel of North American experts to establish a set of competencies in HL for HCP. Subsequent studies showed that most of these listed competencies could be applicable in European countries [14], Chinese [2] or in specific professions, such as nursing [5]. However, the removal and addition of items that occurred in these studies made clear the need to reapply and adapt the skills proposed by the original instrument to other places and cultures.

The establishment of consensus on HL competencies is supported by the new roles expected for HCP as health promoters in clinical settings, as professionals and researchers, according to the Shanghai Declaration on Health Promotion [15]. This declaration focuses on promoting HL, linking the capacity of individuals and communities, as well as the capacity of professionals and health systems to respond to this demand. In addition, the theme is aligned with the need for research in communication and health information provided for in the Agenda of Research Priorities in Brazil [16].

.

Reviewer #2: Comments to the Authors

Method section Line 70...you have included researches done in three languages (English, Spanish and Portuguese). Why only this three languages....?

Authors' response:

The choice of the English language considered the fact that most publications on the subject are in that language [2, 3, 14]. We consider that Brazilian authors publish mostly in Portuguese, English and Spanish. Thus, including Portuguese and Spanish increased the possibility of considering national publications in the composition of the list of competencies in health literacy.

Reviewer #2: Comments to the Authors

Since you intention is to develop competencies in HL could you think that considering six month of professional experience is enough to select participants... Can we say that health professional with six months of experience as an expert. Can you define operationally what we mean an expert. Since the competence's you developed are going to be the part of curricula it needs high experts…

Authors' response:

As described in our manuscript in lines 131-139, the initial identification of health literacy specialists was performed using the criteria of scientific publications on health literacy in Brazil. We used the e-mail address of the corresponding author as a reference, who generally assumes greater responsibility for the published study. Furthermore, the authorship criterion is commonly used in Delphi studies. [4] and replicated in a study similar to ours [5].

In addition, as described in lines 140-147, participants who returned to the invitation email filled out an online form that allowed them to apply the inclusion and exclusion criteria: having a degree in the health area; have experience in direct patient, family or community care, have at least one HL publication in the last five years. Exclusion criteria were: being in function deviation; being away from work activities for more than five years; have exercised only administrative function in the health area.

Thus, in addition to the scientific publication criteria, we are concerned with identifying the experience of our specialists in health services. We have professionals from all regions of the country and 60.0% of the participants had more than six years of experience in care. A previous study showed that the time of six months of experience was enough for the specialists to be able to contribute to the list of competence in health literacy in the Chinese context [2].

Finally, the selection of our experts followed the principle of the power of information [6]. In accordance with this 'principle, our health literacy experts met the objectives of the study, contained specific knowledge about SL, the selection criteria were based on an established theory [4] and our analysis strategy considered the reflection of the group for the consideration of the items, reducing potential individual biases [7]. All participants marked the online form that they considered themselves experts in health literacy.

Reviewer #2: Comments to the Authors

Its better if you move line 101-105 of method section to result part.

Authors' response:

We made the requested change in the manuscript and marked it in red.

Reviewer #2: Comments to the Authors

Method section line 166. Do you have any references to include >= 90% as a cut point.

Authors' response:

There is no consensus on the standard for the level of agreement between authors in the e-Delphi methodology, with a variation of 51.0% - 80.0% in the literature [7]. For the purposes of this study, the value of 90.0% was adopted to define agreement among experts to achieve a high level of consensus and increase the credibility of the study [5]. The cut-off point of 90.0% allowed: increasing the sensitivity of the participants' choice, reducing the risk of entering redundant items and increasing the possibility of including items that are really relevant and feasible to be applied to reality.

Reviewer #2: Comments to the Authors

In discussion section you used other language than english.....Em contraponto, o conhecimento teórico do HL não foi priorizado pelos nossos especialistas refletindo uma proposta prática para os itens de competência. Please could you write it in English,,,as long as you used English as a primary language in this study.

Authors' response:

We made the correction to standardize the language of the manuscript in English.

REFERENCES

1. Cesar FCR, Moraes KL, Brasil VV, Alves AG, Barbosa MA, Oliveira L. Professional Responsiveness to Health Literacy: A Scoping Review. Health literacy research and practice. 2022;6(2):e96-e103. Epub 20220506. doi: 10.3928/24748307-20220418-02. PubMed PMID: 35522856.

2. Chang LC, Chen YC, Wu FL, Liao LL. Exploring health literacy competencies towards patient education programme for Chinese-speaking healthcare professionals: a Delphi study. BMJ open [Internet]. 2017 18 Maio 2019 [cited 2020 jun 10]; 7(1):[e011772 p.]. Available from: http://dx.doi.org/10.1136/bmjopen-2016-011772.

3. Coleman CA, Hudson S, Maine LL. Health literacy practices and educational competencies for health professionals: a consensus study. J Health Commun [Internet]. 2013 18 Maio 2019 [cited 2020 jun 10]; 18(Suppl 1):[82-102 pp.]. Available from: https://doi.org/10.1080/10810730.2013.829538.

4. Baker J, Lovell K, Harris N. How expert are the experts? An exploration of the concept of 'expert' within Delphi panel techniques. Nurse Res [Internet]. 2006 18 Maio 2019 [cited 2020 jun 10]; 14(1):[59-70 pp.]. Available from: https://journals.rcni.com/doi/abs/10.7748/nr2006.10.14.1.59.c6010.

5. Toronto CE. Health literacy competencies for registered nurses: an e-Delphi study. J Contin Educ Nurs [Internet]. 2016 18 Maio 2019 [cited 2020 jun 10]; 47(12):[558-65 pp.]. Available from: https://doi.org/10.3928/00220124-20161115-09.

6. Malterud K, Siersma VD, Guassora AD. Sample size in qualitative interview studies: guided by information power. Qual Health Res [Internet]. 2016 [cited 2020 jun 10]; 26(13):[1753-60 pp.]. Available from: https://doi.org/10.1177/1049732315617444.

7. Keeney S, Hasson F, McKenna HP. The Delphi technique in nursing and health research. United Kingdom: Wiley Online Library; 2011 [cited 2020 jun 10]. Available from: https://onlinelibrary.wiley.com/doi/book/10.1002/9781444392029.

8. Lincoln YS, Guba EG. Naturalistic inquiry. 1st ed. Newbury Park: Sage; 1985. 329 p.

9. Fan ZY, Yang Y, Zhang F. Association between health literacy and mortality: a systematic review and meta-analysis. Arch Public Health [Internet]. 2021 [cited 2021 Jul 21]; 79(1):[119 p.]. Available from: https://doi.org/10.1186/s13690-021-00648-7.

10. Suhail M, Saeed H, Saleem Z, Younas S, Hashmi FK, Rasool F, et al. Association of health literacy and medication adherence with health-related quality of life (HRQoL) in patients with ischemic heart disease. Health Qual Life Outcomes [Internet]. 2021 Apr 13 [cited 2021 Jul 21]; 19(1):[118 p.]. Available from: https://doi.org/10.1186/s12955-021-01761-5.

11. Zheng M, Jin H. The relationship between health literacy and quality of life: a systematic review and meta-analysis. Health Qual Life Out [Internet]. 2018 18 Maio 2019 [cited 2020 jun 10]; 16(1):[201 p.]. Available from: https://doi.org/10.1186/s12955-018-1031-7.

12. Saeed H, Saleem Z, Naeem R, Shahzadi I, Islam M. Impact of health literacy on diabetes outcomes: a cross-sectional study from Lahore, Pakistan. Public health [Internet]. 2018 18 Maio 2019 [cited 2020 jun 10]; 156(1):[8-14 pp.]. Available from: https://doi.org/10.1016/j.puhe.2017.12.005.

13. Morley CM, Levin SA. Health literacy, health confidence, and simulation: a novel approach to patient education to reduce readmissions. Prof Case Manag [Internet]. 2021 May-Jun 01 [cited 2021 Jul 21]; 26(3):[138-49 pp.]. Available from: https://www.ingentaconnect.com/content/wk/ncm/2021/00000026/00000003/art00005;jsessionid=1h0rnorog38le.x-ic-live-03.

14. Karuranga S, Sørensen K, Coleman C, Mahmud AJ. Health literacy competencies for european health care personnel. Health Lit Res Pract [Internet]. 2017 18 Maio 2019 [cited 2020 jun 10]; 1(4):[e247-e56 pp.]. Available from: https://doi.org/10.3928/24748307-20171005-01.

15. Organização Mundial da Saúde. Shanghai declaration on health promotion. China: OMS; 2016 [cited 2020 jun 10]. Available from: https://www.who.int/healthpromotion/conferences/9gchp/shanghai-declaration.pdf?ua=1.

16. Ministério da Saúde. Agenda de Prioridades de Pesquisa do Ministério da Saúde - APPMS. Brasília: Ministério da Saúde; 2018 [cited 2021 jun 29]. Available from: https://bvsms.saude.gov.br/bvs/publicacoes/agenda_prioridades_pesquisa_ms.pdf.

---

## [Editor Report · Decision Letter 1]

29 Jun 2022

COMPETENCIES OF HEALTH PERSONNEL FOR THE PRACTICE OF HEALTH LITERACY IN BRAZIL: A DELPHI CONSENSUS SURVEY

PONE-D-22-06015R1

Dear Dr. Cesar,

We’re pleased to inform you that your manuscript has been judged scientifically suitable for publication and will be formally accepted for publication once it meets all outstanding technical requirements.

Kind regards,

Amene Abebe Kerbo, Ph.D.

Academic Editor

PLOS ONE
---

## [Editor Report · Acceptance letter]

12 Jul 2022

PONE-D-22-06015R1 

Competencies of health personnel for the practice of health literacy in Brazil: a Delphi consensus survey 

Dear Dr. Cesar:

I'm pleased to inform you that your manuscript has been deemed suitable for publication in PLOS ONE. Congratulations! Your manuscript is now with our production department. 

Kind regards, 

on behalf of

Dr. Amene Abebe Kerbo 

Academic Editor

PLOS ONE